# Understanding Smoothness of Vector Gaussian Processes on Product Spaces

**Emilio Porcu**  *emilio.porcu@ku.ac.ae*
*Department of Mathematics, College of Computing and Mathematical Sciences*
*Khalifa University, Abu Dhabi*
*& ADIA Lab, Abu Dhabi*

**Ana Paula Peron**  *apperon@icmc.usp.br*
*Departamento de Matemática, Instituto de Ciências Matemáticas e de Computação,*
*Universidade de São Paulo, São Carlos SP, Brazil.*

**Eugenio Massa**  *eug.massa@gmail.com*
*Departamento de Matemática, Instituto de Ciências Matemáticas e de Computação,*
*Universidade de São Paulo, São Carlos SP, Brazil.*

**Xavier Emery**  *xemery@ing.uchile.cl*
*Department of Mining Engineering, Universidad de Chile*
*& Advanced Mining Technology Center, Universidad de Chile*

**Reviewed on OpenReview:** *https://openreview.net/forum?id=XXXX*

## Abstract

Vector Gaussian processes are becoming increasingly important in machine learning and statistics, with applications to many branches of applied sciences. Recent efforts have allowed to understand smoothness in scalar Gaussian processes defined over manifolds as well as over product spaces involving manifolds.

Under assumptions of Gaussianity and mean-square continuity, the smoothness of a zero-mean scalar process is in one-to-one correspondence with the smoothness of the covariance kernel. Unfortunately, such a result is not available for vector-valued random fields, as the way each component in the covariance kernel contributes to the smoothness of the vector field is unclear.

This paper challenges the problem of quantifying smoothness of matrix-valued continuous kernels that are associated with mean-square continuous vector Gaussian processes defined over non-Euclidean product manifolds. After noting that a constructive RKHS approach is unsuitable for this specific task, we proceed through the analysis of spectral properties. Specifically, we find a spectral representation to quantify smoothness through Sobolev spaces that are adapted to certain measure spaces of product measures obtained through the tensor product of Haar measures with multivariate Gaussian measures. Our results allow to measure smoothness in a simple way, and open for the study of foundational properties of certain machine learning techniques over product spaces.

## 1 Introduction

### 1.1 Context

The paper deals with the smoothness of continuous matrix-valued kernels associated with mean-square continuous vector-valued Gaussian processes defined on the product of two spaces, with one of them being non-Euclidean, namely a hypersphere of $d$ dimensions embedded in a $(d+1)$-dimensional Euclidean space.

Gaussian processes (Seeger, 2004) are ubiquitous in machine learning, statistics and numerical analysis. Vector (i.e., multivariate) Gaussian processes have recently received attention after the constructive approaches proposed by Hutchinson et al. (2021). The impact of such processes on the machine learning community ranges from regression (Chen et al., 2023), Bayesian optimization and active learning (see the discussion in Hutchinson et al., 2021, and references therein), to relevance vector machines (Quinonero-Candela, 2004), sensor networks (Osborne et al., 2008), text categorization (Kazawa et al., 2005), informance vector machines (Lawrence et al., 2002), gradual learning (Yuan et al., 2022), and multitask learning (Bonilla et al., 2007; Xing et al., 2021).

Vector Gaussian processes arise within the framework of multiple output learning of a vector-valued function $\boldsymbol{f} = (f_1, \ldots, f_p)^\top$ that is observed over a finite set $\boldsymbol{Y} = \boldsymbol{f}(\boldsymbol{X}) := \{\boldsymbol{f}(\underline{\boldsymbol{x}}_1), \ldots, \boldsymbol{f}(\underline{\boldsymbol{x}}_N)\}$, from training data $\underline{\boldsymbol{x}}$ collected over the training set $\boldsymbol{X} = \{\underline{\boldsymbol{x}}_1, \ldots, \underline{\boldsymbol{x}}_N\}$. Specifically, we suppose that $\underline{\boldsymbol{x}}$ is defined over a product space $\Upsilon^{(d,k)} = \mathbb{S}^d \times \mathbb{R}^k$, with $\mathbb{S}^d$ being the unit sphere of dimension $d$ and $\mathbb{R}^k$ being the $k$-dimensional Euclidean space. The output space $\mathbb{Y}$, where $\boldsymbol{f}$ is defined, has dimension $p$.

The problem can be tackled either assuming that $\boldsymbol{f}$ belongs to a reproducing kernel Hilbert space (RKHS) of vector-valued functions or assuming that $\boldsymbol{f}$ is drawn from a vector Gaussian process.

We start by illustrating the RKHS perspective for vector-valued functions that are reproduced through matrix-valued kernels, denoted $\widetilde{\boldsymbol{K}}$ throughout, being matrix-valued functions from $\Upsilon^{(d,k)} \times \Upsilon^{(d,k)}$ into $\mathbb{R}^{p \times p}$. A vector RKHS is a Hilbert space, $\mathcal{H}_{\widetilde{\boldsymbol{K}}}$, composed of vector-valued functions $\boldsymbol{f}$ such that, for all $\boldsymbol{c} \in \mathbb{R}^p$ and all $\underline{\boldsymbol{x}} \in \Upsilon^{(d,k)}$, the linear combination $\boldsymbol{f}(\underline{\boldsymbol{x}})^\top \boldsymbol{c}$ is obtained through

$$\boldsymbol{f}(\underline{\boldsymbol{x}})^\top \boldsymbol{c} = \langle \boldsymbol{f}(\cdot), \widetilde{\boldsymbol{K}}(\cdot, \underline{\boldsymbol{x}}) \boldsymbol{c} \rangle_{\mathcal{H}_{\widetilde{\boldsymbol{K}}}}, \tag{1}$$

where $\langle \cdot, \cdot \rangle_{\mathcal{H}_{\widetilde{\boldsymbol{K}}}}$ is the inner product on $\mathcal{H}_{\widetilde{\boldsymbol{K}}}$. The kernel $\widetilde{\boldsymbol{K}}$ is positive semidefinite: for any arbitrary system $\{\boldsymbol{c}_k\}_{k=1}^N$ of $p$-dimensional real vectors and any finite collection of points $\{\underline{\boldsymbol{x}}_k\}_{k=1}^N$ in the input space, we have

$$\sum_{h=1}^N \sum_{k=1}^N \boldsymbol{c}_h^\top \widetilde{\boldsymbol{K}}(\underline{\boldsymbol{x}}_h, \underline{\boldsymbol{x}}_k) \boldsymbol{c}_k \geq 0.$$

For a thorough review about RKHS for both scalar and vector-valued settings, as well for material about regularization in RKHS, the reader is referred to Hofmann et al. (2008) and Alvarez et al. (2012).

Although RKHS are a powerful instrument to quantify smoothness of scalar Gaussian processes, the same does not hold for the case of vector Gaussian processes, where the role of the matrix-valued kernel remains unclear. Hence, we opt here for the alternative of matrix-valued kernels through vector-valued Gaussian processes.

## 1.2 Why Study Smoothness? Why Product Spaces?

Smoothness plays a fundamental role in numerous applications to machine learning and statistics. We mention here the most recent development in Gaussian process regression. The recent work by Rosa et al. (2023) deals with Bayesian contraction rates under the framework of Gaussian process regression with random design. Posterior construction rates provide a nice way to illustrate how a given class of posteriors concentrates around the true data generating process. Rosa et al. (2023) prove that the contraction rates depend on the smoothness of the underlying Gaussian process, the prior of which is defined through a Matérn kernel (Porcu et al., 2023).

Well-known results from probability theory connect the smoothness properties of a scalar Gaussian process with those of the associated kernel (Yadrenko, 1983; Adler and Taylor, 2007). Unfortunately, these results are not available for the vector-valued case, and the same definition of RKHS as in Equation (1) clearly shows that the contribution of each component in $\widetilde{\boldsymbol{K}}$ to the smoothness of $\boldsymbol{Z}$ is unclear. This explains the lack of literature in this direction and the fact that the available contributions are centered to the smoothness of the kernel $\widetilde{\boldsymbol{K}}$ rather than the smoothness of $\boldsymbol{Z}$.

An intuitive way to look at the geometric smoothness properties of the kernel is by working under the framework of Sobolev spaces.

Another relevant motivation for studying smoothness of Gaussian processes on manifolds is related to the use of computational tools of *kernel cubature* and *kernel discrepancy* beyond the usual Euclidean manifold. Barp et al. (2022) illustrate the importance of Sobolev spaces when quantifying kernel cubatures. These topics have been popular in statistics, machine learning, and numerical analysis. Kernel cubature has been applied in several contexts, and the reader is referred to Hubbert et al. (2023), with the references therein.

Studying smoothness on non-Euclidean manifolds has been important to several disciplines. For the special case of the manifold being a *d*-dimensional sphere, applications include kernel cubature (Marques et al., 2013; 2022), Stein's method to numerically calculate posterior expectations in directional statistics (Barp et al., 2022), and approximation of solutions of some classes of PDEs (see e.g. Fasshauer, 2007). Not to mention that certain classes of kernels on spheres ensure that the solution of the PDE belongs to the RKHS and, through the use of an appropriate kernel method, can be consistently approximated (see Fuselier and Wright, 2009; 2012; Hubbert et al., 2015).

The product space $\Upsilon^{(d,k)}$ has received increasing attention in the statistics and machine learning communities (Atluri et al., 2018; Wang et al., 2022; Porcu et al., 2016; 2021). Applications from several branches of science justify this context, such as

- atmospheric, environmental and oceanographic sciences: variables such as air pressure, air humidity, wind speed, surface temperature, solar radiation, aerosol optical depth, daily ozone concentration, ground level concentration of particulate matter, ocean current velocity, sea surface height anomaly, sea surface salinity, sea surface chlorophyll-a concentration, or sea water density are indexed by the position (longitude and latitude) on planet Earth ($\mathbb{S}^2$) and by time ($\mathbb{R}^1$) (Castruccio and Stein, 2013; Oleson et al., 2013; Faghmous and Kumar, 2014; Jeong et al., 2017; Wu et al., 2022);

- geophysics: seismic and volcanic events can be represented by point processes indexed by a position on planet Earth ($\mathbb{S}^2$) and by time ($\mathbb{R}^1$) (Illian et al., 2008; Connor et al., 2009);

- structural geology and geotechnics: variables such as the linear discontinuity frequency, the rock quality designation, or the uniaxial compressive strength, used to measure the geotechnical quality of a rock mass, are indexed by the position (easting, northing, elevation, i.e., a point of $\mathbb{R}^3$) of the rock core sample in the rock mass and by the orientation (azimuth and dip, i.e., a point of $\mathbb{S}^2$) of this sample (Sánchez et al., 2019; 2021);

- social science: crime and security data can be indexed by their position in the geographic space ($\mathbb{R}^2$) and by time at the scale of a day ($\mathbb{S}^1$) (Tompson et al., 2015; Shirota and Gelfand, 2017);

- neuroscience: functional magnetic resonance imaging (fMRI), electroencephalography (EEG) and magnetoencephalography (MEG) signals are indexed by the position on the human head ($\mathbb{S}^2$) and by time ($\mathbb{R}^1$) (Wingeier et al., 2001; Atluri et al., 2016).

### 1.3 Challenges and Contribution

The smoothness of scalar Gaussian processes was studied in Lang and Schwab (2013) for the sphere, and in Clarke De la Cerda et al. (2018) for the product of a sphere with $\mathbb{R}$.

While scalar Gaussian processes are well understood, the literature on smoothness of matrix-valued kernels in machine learning is scarce, with the notable exception of Cleanthous (2023), who provides an ingenious construction for a Gaussian process defined over a ball embedded in $\mathbb{R}^k$.

Our paper contributes in this direction. Specifically,
**a)** Section 2 considers mean-square continuous zero-mean vector Gaussian processes defined over the space $\Upsilon^{(d,k)}$ as being previously defined;
**b)** we take a spectral path to smoothness of their covariance kernels in Section 3, through a proper spectral representation for a vector Gaussian process on $\Upsilon^{(d,k)}$, and consequently for the related matrix-valued

kernel;

**c)** we construct, in Section 3.3, a suitable Sobolev space for such a matrix-valued kernel;

**d)** Section 3.4 provides a spectral characterization of smoothness of the kernel.

### 1.4 Outline and Notation

Section 2 provides a succinct mathematical background. Section 3 illustrates the way to construct proper Sobolev spaces through spectral representations over the space $\Upsilon^{(d,k)}$. Proofs are technical and deferred to Appendix A. Section 5 concludes the paper with a discussion.

Hereinafter, $\mathbb{Z}$ is the set of integer numbers, $\mathbb{Z}_+ = \{\kappa \in \mathbb{Z} : \kappa \geq 0\}$, i stands for the complex imaginary unit, $p$, $d$ and $k$ for positive integers, and $\|\cdot\|$ for the Euclidean norm on $\mathbb{R}^k$. Bold letters denote vectors or matrices of size $p \times p$. $\overline{\boldsymbol{A}}$ refers to the conjugate of a complex matrix $\boldsymbol{A}$, and $\boldsymbol{A}^\top$ to its transpose. In order to work in multidimensional spaces, we consider the multi-index notation: for $\boldsymbol{\alpha} = (\alpha_1, \ldots, \alpha_k) \in \mathbb{Z}_+^k$ and $\boldsymbol{h} = (h_1, \ldots, h_k) \in \mathbb{R}^k$, we set

$$|\boldsymbol{\alpha}| = \sum_{i=1}^{k} \alpha_i, \qquad \boldsymbol{\alpha}! = \prod_{i=1}^{k} \alpha_i!, \qquad \partial^{\boldsymbol{\alpha}} f(\boldsymbol{h}) = \partial_{h_1}^{\alpha_1} \partial_{h_2}^{\alpha_2} \ldots \partial_{h_k}^{\alpha_k} f(\boldsymbol{h}),$$

and for $\boldsymbol{\alpha}, \boldsymbol{\beta} \in \mathbb{Z}_+^k$

$$\boldsymbol{\alpha}^{\boldsymbol{\beta}} = \prod_{i=1}^{k} \alpha_i^{\beta_i}, \qquad \boldsymbol{\alpha} \geq \boldsymbol{\beta} \iff \alpha_i \geq \beta_i, \ \forall i,$$

with the usual understanding that $0^0 = 1$.

## 2 Vector Gaussian Processes

Let

$$\Upsilon^{(d,k)} := \mathbb{S}^d \times \mathbb{R}^k = \{\underline{\boldsymbol{x}} = (\boldsymbol{x}, \boldsymbol{t}) \in \mathbb{R}^{d+1} \times \mathbb{R}^k : \|\boldsymbol{x}\| = 1, \boldsymbol{t} \in \mathbb{R}^k\}.$$

A $p$-variate (vector) Gaussian process, $\{\boldsymbol{Z}(\underline{\boldsymbol{x}}) : \underline{\boldsymbol{x}} \in \Upsilon^{(d,k)}\}$ is an uncountable collection of random vectors such that, for any finite collection of points $\underline{\boldsymbol{x}}_1, \ldots, \underline{\boldsymbol{x}}_N \in \Upsilon^{(d,k)}$, the vector $(\boldsymbol{Z}(\underline{\boldsymbol{x}}_1)^\top, \ldots, \boldsymbol{Z}(\underline{\boldsymbol{x}}_N)^\top)^\top$, having dimension $(p \times N) \times 1$, is a Gaussian random vector. In what follows, without loss of generality, we suppose that such a Gaussian process has a zero mean at any point $\underline{\boldsymbol{x}}$ in $\Upsilon^{(d,k)}$.

A *$p$-variate covariance kernel* on $\Upsilon^{(d,k)}$ is a matrix-valued function

$$\widetilde{\boldsymbol{K}} : \Upsilon^{(d,k)} \times \Upsilon^{(d,k)} \to \mathbb{R}^{p \times p}$$

defined as

$$\widetilde{\boldsymbol{K}}(\underline{\boldsymbol{x}}, \underline{\boldsymbol{x}}') = [\widetilde{K}_{ij}(\underline{\boldsymbol{x}}, \underline{\boldsymbol{x}}')]_{i,j=1}^{p}, \qquad \underline{\boldsymbol{x}}, \underline{\boldsymbol{x}}' \in \Upsilon^{(d,k)},$$

where $\widetilde{K}_{ij}(\underline{\boldsymbol{x}}, \underline{\boldsymbol{x}}') = \widetilde{K}_{ji}(\underline{\boldsymbol{x}}', \underline{\boldsymbol{x}})$ for all $i, j \in \{1, \ldots, p\}$ and $\underline{\boldsymbol{x}}, \underline{\boldsymbol{x}}' \in \Upsilon^{(d,k)}$, and where $\widetilde{\boldsymbol{K}}$ is positive semidefinite, that is, the $pN \times pN$ block matrix $[\widetilde{\boldsymbol{K}}(\underline{\boldsymbol{x}}_m, \underline{\boldsymbol{x}}_n)]_{m,n=1}^{N}$ is symmetric and nonnegative definite for any set of points $\underline{\boldsymbol{x}}_1, \ldots, \underline{\boldsymbol{x}}_N \in \Upsilon^{(d,k)}$.

Hereinafter, we focus on the case where the mapping $\widetilde{\boldsymbol{K}}$ is continuous, isotropic on $\mathbb{S}^d$ and stationary in $\mathbb{R}^k$, meaning that

$$\widetilde{\boldsymbol{K}}(\underline{\boldsymbol{x}}, \underline{\boldsymbol{x}}') = \boldsymbol{K}(\langle \boldsymbol{x}, \boldsymbol{x}' \rangle, \boldsymbol{t} - \boldsymbol{t}'), \quad \underline{\boldsymbol{x}} = (\boldsymbol{x}, \boldsymbol{t}), \underline{\boldsymbol{x}}' = (\boldsymbol{x}', \boldsymbol{t}'), \tag{2}$$

with $\langle \cdot, \cdot \rangle$ denoting the dot product in $\mathbb{R}^{d+1}$, and $\boldsymbol{K} : [-1, 1] \times \mathbb{R}^k \to \mathbb{R}^{p \times p}$ being a continuous mapping. Throughout, $\boldsymbol{K}$ will be called a kernel for simplicity, albeit this should be called the isotropic profile of the kernel $\widetilde{\boldsymbol{K}}$. This will not give rise to confusion as, from now, only the mapping $\boldsymbol{K}$ will be used.

On the one hand, Alegría et al. (2019, Theorem 6.2) proved that a continuous kernel $\boldsymbol{K}$ of the type (2) can be uniquely decomposed as follows:

$$\boldsymbol{K}(s, \boldsymbol{h}) = \sum_{n=0}^{\infty} \dim(\mathcal{H}_n^d)\, \boldsymbol{C}_n^d(\boldsymbol{h})\, \mathcal{G}_n^{(d-1)/2}(s), \qquad s \in [-1, 1], \quad \boldsymbol{h} \in \mathbb{R}^k, \tag{3}$$

for a sequence $\{\boldsymbol{C}_n^d(\cdot)\}_{n=0}^{\infty}$ of matrix-valued stationary covariance kernels such that $\{\dim(\mathcal{H}_n^d)\boldsymbol{C}_n^d(\boldsymbol{0})\}_{n=0}^{\infty}$ is summable. For $d > 1$, $\mathcal{G}_n^{(d-1)/2}$ is defined in terms of the Gegenbauer (ultraspherical) polynomial $G_n^{(d-1)/2}$, normalizing as $\mathcal{G}_n^{(d-1)/2} = G_n^{(d-1)/2}/G_n^{(d-1)/2}(1)$, while for $d = 1$, $\mathcal{G}_n^0 = T_n$ is the $n$th Chebyshev polynomial of the first kind.

On the other hand, the generalized addition theorem for spherical harmonics (Erdélyi, 1953, Equation 11.4.2) states that

$$\sum_{q \in \mathcal{A}_{n,d}} \mathcal{Y}_{n,q,d}(\boldsymbol{x})\mathcal{Y}_{n',q',d}(\boldsymbol{x}') = \dim(\mathcal{H}_n^d)\, \mathcal{G}_n^{(d-1)/2}(\langle \boldsymbol{x}, \boldsymbol{x}' \rangle), \quad \boldsymbol{x}, \boldsymbol{x}' \in \mathbb{S}^d, \quad n \in \mathbb{Z}_+, \tag{4}$$

where $\mathcal{A}_{n,d}$ is a set of finite cardinality $\dim(\mathcal{H}_n^d)$ associated with the spherical harmonics $\mathcal{Y}_{n,q,d}$, which form an orthonormal basis for all Lebesgue-square-integrable measurable functions on the $d$-dimensional sphere $\mathbb{S}^d$.

Owing to Equations (3) and (4), the Karhunen theorem on the generalized spectral representation of random processes (Yaglom, 1987b, Equation (2.31')) allows decomposing a mean-square continuous $p$-variate Gaussian process $\boldsymbol{Z}$ having a zero mean and $\boldsymbol{K}$ as its covariance kernel in the following fashion:

$$\boldsymbol{Z}(\underline{\boldsymbol{x}}) = \sum_{n=0}^{\infty} \sum_{q \in \mathcal{A}_{n,d}} \boldsymbol{A}_{n,q}^d(\boldsymbol{t})\mathcal{Y}_{n,q,d}(\boldsymbol{x}), \qquad \underline{\boldsymbol{x}} = (\boldsymbol{x}, \boldsymbol{t}) \in \Upsilon^{(d,k)}, \tag{5}$$

where $\{\boldsymbol{A}_{n,q}^d(\cdot)\}$ is a sequence of mean-square continuous zero-mean vector Gaussian processes in $\mathbb{R}^k$ such that (`cov` refers to the covariance operator)

$$\mathrm{cov}\left(\boldsymbol{A}_{n,q}^d(\boldsymbol{t}), \boldsymbol{A}_{n',q'}^d(\boldsymbol{t}')\right) = \delta_{n,n'}\, \delta_{q,q'}\, \boldsymbol{C}_n^d(\boldsymbol{t} - \boldsymbol{t}'), \quad \boldsymbol{t}, \boldsymbol{t}' \in \mathbb{R}^k, \quad n, n' \in \mathbb{Z}_+, \quad q \in \mathcal{A}_{n,d}, \; q' \in \mathcal{A}_{n',d}. \tag{6}$$

The reciprocal, that a vector Gaussian process of the form (5) has a covariance kernel of the form (3), has been established in the scalar case ($p = 1$) by Clarke De la Cerda et al. (2018, Proposition 3.1). For the reader's convenience, we sketch the proof for the vector case. For any $\underline{\boldsymbol{x}} = (\boldsymbol{x}, \boldsymbol{t})$ and $\underline{\boldsymbol{x}}' = (\boldsymbol{x}', \boldsymbol{t}')$ in $\Upsilon^{(d,k)}$, the bilinearity of the covariance implies

$$
\begin{aligned}
\widetilde{\boldsymbol{K}}(\underline{\boldsymbol{x}}, \underline{\boldsymbol{x}}') &= \mathrm{cov}\left(\boldsymbol{Z}(\underline{\boldsymbol{x}}), \boldsymbol{Z}(\underline{\boldsymbol{x}}')\right) \\
&= \mathrm{cov}\left(\sum_{n=0}^{\infty} \sum_{q \in \mathcal{A}_{n,d}} \boldsymbol{A}_{n,q}^d(\boldsymbol{t})\mathcal{Y}_{n,q,d}(\boldsymbol{x}), \sum_{n'=0}^{\infty} \sum_{q' \in \mathcal{A}_{n',d}} \boldsymbol{A}_{n',q'}^d(\boldsymbol{t}')\mathcal{Y}_{n',q',d}(\boldsymbol{x}')\right) \\
&= \sum_{n=0}^{\infty} \sum_{q \in \mathcal{A}_{n,d}} \sum_{n'=0}^{\infty} \sum_{q' \in \mathcal{A}_{n',d}} \mathcal{Y}_{n,q,d}(\boldsymbol{x})\mathcal{Y}_{n',q',d}(\boldsymbol{x}')\, \mathrm{cov}\left(\boldsymbol{A}_{n,q}^d(\boldsymbol{t}), \boldsymbol{A}_{n',q'}^d(\boldsymbol{t}')\right) \\
&= \sum_{n=0}^{\infty} \left(\sum_{q \in \mathcal{A}_{n,d}} \mathcal{Y}_{n,q,d}(\boldsymbol{x})\mathcal{Y}_{n',q',d}(\boldsymbol{x}')\right) \boldsymbol{C}_n^d(\boldsymbol{t} - \boldsymbol{t}') \\
&= \sum_{n=0}^{\infty} \dim(\mathcal{H}_n^d)\, \boldsymbol{C}_n^d(\boldsymbol{t} - \boldsymbol{t}')\, \mathcal{G}_n^{(d-1)/2}(\langle \boldsymbol{x}, \boldsymbol{x}' \rangle),
\end{aligned}
$$

with the last two equalities derived from Equations (6) and (4).

Equation (5) can be coupled with the spectral representation of the mean-square continuous stationary random process $\boldsymbol{A}_{n,q}^d(\cdot)$ (see Yaglom (1987a, Equation (4.70)) or Chilès and Delfiner (2012, Equation (2.16)))

to attain

$$\boldsymbol{Z}(\underline{\boldsymbol{x}}) = \sum_{n=0}^{\infty} \sum_{q \in \mathcal{A}_{n,d}} \int_{\mathbb{R}^k} e^{i\langle \boldsymbol{t}, \boldsymbol{\omega} \rangle} \boldsymbol{\xi}_{n,q}(d\boldsymbol{\omega}) \mathcal{Y}_{n,q,d}(\boldsymbol{x}), \qquad \underline{\boldsymbol{x}} = (\boldsymbol{x}, \boldsymbol{t}) \in \Upsilon^{(d,k)}, \tag{7}$$

where $\{\boldsymbol{\xi}_{n,q}(d\cdot)\}_{n,q}^{\infty}$ is a sequence of vector-valued measures with orthogonal increments, that is, $\mathbb{E}\left(\boldsymbol{\xi}_{n,q}(A)\overline{\boldsymbol{\xi}_{n',q}(B)}\right) = \delta_{n=n'}\boldsymbol{F}_n(A\bigcap B)$, for all $q$, $n$, $n'$ and all Borel sets $A$ and $B$ in $\mathbb{R}^k$, where $\boldsymbol{F}_n$ is a matrix-valued measure of bounded variation such that $\boldsymbol{F}_n(d\boldsymbol{\omega})$ is a positive semidefinite matrix for all $\boldsymbol{\omega} \in \mathbb{R}^k$.

Under the additional condition $\sum_n \dim(\mathcal{H}_n^d) \int_{\mathbb{R}^k} \boldsymbol{\xi}_{n,\zeta}(d\boldsymbol{\omega}) < \infty$, Equation (3) becomes

$$\boldsymbol{K}(s, \boldsymbol{h}) = \sum_{n=0}^{\infty} \dim(\mathcal{H}_n^d)\left(\int_{\mathbb{R}^k} e^{i\langle \boldsymbol{h}, \boldsymbol{\omega} \rangle} \boldsymbol{F}_n(d\boldsymbol{\omega})\right) \mathcal{G}_n^{(d-1)/2}(s), \qquad s \in [-1, 1], \quad \boldsymbol{h} \in \mathbb{R}^k. \tag{8}$$

Clearly, $\boldsymbol{C}_n^d$ is real matrix-valued if, and only if, $\boldsymbol{F}_n(A) = \overline{\boldsymbol{F}_n(-A)}$, for all Borel sets $A$ in $\mathbb{R}^k$. A stronger condition for this to happen is that $\boldsymbol{\xi}_n(-A) = \boldsymbol{\xi}_n(A)^\top$. Throughout, we shall always work under the assumption of real matrix-valued covariance kernels.

## 3 Understanding Regularities of Matrix-Valued Kernels

To study the properties of the matrix-valued kernel $\boldsymbol{K}$, an intuitive approach is to provide a Karhunen-Loève expansion of a vector Gaussian process in $\mathbb{R}^p$, with input space $\Upsilon^{(d,k)}$. Since the product space $\Upsilon^{(d,k)}$ is not compact, an extension of the arguments provided for the scalar case by Clarke De la Cerda et al. (2018) suggests that a sensible strategy is needed to provide a legitimate Karhunen-Loève expansion of vector-valued functions. There are indeed two possibilities:

**a)** to *compactify* the space $\Upsilon^{(d,k)}$ by considering the space $\Upsilon_T^{(d,k)} := \mathbb{S}^d \times [0, T]^k$, with $T$ a positive constant. Under such a construction, a Karhunen-Loève expansion can be namely obtained. Yet, this approach has a cost in that it does not allow for traditional spectral expansions as much as in Equations (7) and (8), respectively;

**b)** to consider the measure space

$$\left(\Upsilon^{(d,k)}, \mathbb{B}, \mu_{\Upsilon^{(d,k)}}\right), \tag{9}$$

where $\mathbb{B}$ is the Borel sigma-algebra over $\Upsilon^{(d,k)}$, and where $\mu_{\Upsilon^{(d,k)}}$ is a product measure defined through

$$\mu_{\Upsilon^{(d,k)}}(d\underline{\boldsymbol{x}}) = \sigma_d(d\boldsymbol{x}) \times \boldsymbol{\nu}(d\boldsymbol{t}), \qquad \underline{\boldsymbol{x}} \in \Upsilon^{(d,k)},$$

where $\sigma_d$ is the Haar measure, i.e., the Lebesgue measure for the sphere, and $\boldsymbol{\nu}$ is the Gaussian measure in $\mathbb{R}^k$ with zero-vector mean and identity covariance matrix, i.e.,

$$\boldsymbol{\nu}(d\boldsymbol{t}) = (2\pi)^{-k/2} e^{-\|\boldsymbol{t}\|^2/2} d\boldsymbol{t}. \tag{10}$$

Under this choice, the Karhunen-Loève expansion for the vector Gaussian process $\boldsymbol{Z}$ can be attained at the expense of defining a suitable orthonormal basis that is legitimate for this measure space. Our paper takes this path. Hence, we start by defining a proper orthonormal basis for the case considered here.

We illustrate our routine through the following scheme.

> **The Route to Smoothness**
>
> 1. Consider the measure space in Equation (9).
>
> 2. Provide an orthonormal basis.
>
> 3. Provide a suitable Karhunen-Loève expansion.
>
> 4. Define a proper Sobolev space.
>
> 5. Quantify smoothness of the kernel $\boldsymbol{K}$.

The following sections detail each of the steps in this routine.

## 3.1 A Constructive Approach to Orthonormal Bases

Consider the normalized Hermite polynomials $H_\kappa$ on the real line defined by (Olver et al., 2010, Table 18.3.1)

$$H_\kappa(\xi) = \frac{(-1)^\kappa}{(\kappa!)^{1/2}} e^{\frac{\xi^2}{2}} \frac{\mathrm{d}^\kappa}{\mathrm{d}\xi^\kappa} e^{\frac{-\xi^2}{2}}, \quad \xi \in \mathbb{R}, \quad \kappa = 0, 1, 2, \ldots.$$

The family $\{H_\kappa\}_{\kappa \in \mathbb{Z}_+}$ forms a complete orthonormal system for $L^2(\mathbb{R}, \nu)$, with the standard Gaussian measure $\mathrm{d}\nu = (2\pi)^{-1/2} e^{-\xi^2/2} \mathrm{d}\xi$, i.e.,

$$\frac{1}{\sqrt{2\pi}} \int_{-\infty}^{\infty} H_\kappa(\xi) H_{\kappa'}(\xi) e^{\frac{-\xi^2}{2}} \mathrm{d}\xi = \delta_{\kappa,\kappa'}. \tag{11}$$

Moreover, the $l$-th derivative of the Hermite polynomials satisfies

$$\frac{\mathrm{d}^l}{\mathrm{d}\xi^l} H_\kappa(\xi) = \sqrt{\frac{\kappa!}{(\kappa - l)!}} \, H_{\kappa - l}(\xi). \tag{12}$$

On $\mathbb{R}^k$, $k \geq 2$, we define the normalized multiple Hermite functions $\Phi_{\boldsymbol{\alpha}}$, with $\boldsymbol{\alpha} \in \mathbb{Z}_+^k$ through the identity

$$\Phi_{\boldsymbol{\alpha}}(\boldsymbol{h}) = \prod_{i=1}^{k} H_{\alpha_i}(h_i), \quad \boldsymbol{h} \in \mathbb{R}^k. \tag{13}$$

It can be verified via Fubini's theorem, by Equation (11) and the definition of $\boldsymbol{\nu}$ in Equation (10), that these functions form an orthonormal basis of $L^2(\mathbb{R}^k, \boldsymbol{\nu})$.

Hence, we have completed Step 2 in our *Route to Smoothness*.

## 3.2 Expansion for the Matrix-Valued Kernel

The sequence $\{\boldsymbol{C}_n(\cdot)\}_{n=0}^{\infty}$ in the series expansion (3) is summable at zero. Further, from well-known properties of matrix-valued positive semidefinite functions (Chilès and Delfiner, 2012), we have that, for every $n = 0, 1, \ldots$, the matrix-valued function $\boldsymbol{C}_n$ having elements $C_{ij,n}$, $i, j = 1, \ldots, p$, satisfies $C_{ij,n}(\boldsymbol{h})^2 \leq C_{ii,n}(\boldsymbol{0}) C_{jj,n}(\boldsymbol{0})$, for $i, j = 1, \ldots, p$. This in turn implies that, for every finite measure $\boldsymbol{\lambda}$, the mapping $\boldsymbol{C}_n$ is in $L^2(\mathbb{R}^k, \boldsymbol{\lambda})$ for all $n$. This is obviously true when $\boldsymbol{\lambda}$ is the Gaussian measure $\boldsymbol{\nu}$.

From Equation (3) in concert with the fact that

$$\left| \mathcal{G}_n^{(d-1)/2}(s) \right| = \left| \frac{G_n^{(d-1)/2}(s)}{G_n^{(d-1)/2}(1)} \right| \leq 1, \quad s \in [-1, 1],$$

we conclude that the convergence of the series (3) is uniform.

At this point, since the multivariate Hermite polynomials that have been defined in Equation (13) form a complete orthonormal basis in $L^2(\mathbb{R}^k, \boldsymbol{\nu})$, we have that, for every $n = 0, 1, \ldots$, the positive semidefinite functions $\boldsymbol{C}_n : \mathbb{R}^k \to \mathbb{R}^{p \times p}$ can be uniquely expanded in terms of Hermite polynomials, that is,

$$\boldsymbol{C}_n^d(\boldsymbol{h}) = \sum_{\boldsymbol{\alpha} \in \mathbb{Z}_+^k} \boldsymbol{\gamma}_{n,\boldsymbol{\alpha}}^d \Phi_{\boldsymbol{\alpha}}(\boldsymbol{h}), \quad \boldsymbol{h} \in \mathbb{R}^k,$$

where the series converges in $L^2(\mathbb{R}^k, \boldsymbol{\nu})$, and where $\left\{\boldsymbol{\gamma}_{n,\boldsymbol{\alpha}}^d\right\}_{\boldsymbol{\alpha} \in \mathbb{Z}_+^k} \subset \mathbb{C}^{p \times p}$ is a summable sequence of matrices such that

$$\boldsymbol{\gamma}_{n,\boldsymbol{\alpha}}^d = \int_{\mathbb{R}^k} \boldsymbol{C}_n^d(\boldsymbol{h}) \Phi_{\boldsymbol{\alpha}}(\boldsymbol{h}) \, \boldsymbol{\nu}(\mathrm{d}\boldsymbol{h}), \quad n \in \mathbb{Z}_+, \quad \boldsymbol{\alpha} \in \mathbb{Z}_+^k. \tag{14}$$

Consequently, the kernel $\boldsymbol{K}$ in Equation (3) can be uniquely expanded as

$$\boldsymbol{K}(s, \boldsymbol{h}) = \sum_{n=0}^{\infty} \dim(\mathcal{H}_n^d) \sum_{\boldsymbol{\alpha} \in \mathbb{Z}_+^k} \boldsymbol{\gamma}_{n,\boldsymbol{\alpha}}^d \Phi_{\boldsymbol{\alpha}}(\boldsymbol{h}) \mathcal{G}_n^{(d-1)/2}(s), \quad (s, \boldsymbol{h}) \in [-1, 1] \times \mathbb{R}^k. \tag{15}$$

We call the indexed set $\left\{\boldsymbol{\gamma}_{n,\boldsymbol{\alpha}}^d\right\}_{(n,\boldsymbol{\alpha}) \in \Omega_k} \subset \mathbb{C}^{p \times p}$ the *Gegenbauer-Hermite spectrum* of the $p$-variate kernel $\boldsymbol{K}$, where the indices take value in the set

$$\Omega_k := \{(n, \boldsymbol{\alpha}) : \ n \in \mathbb{Z}_+, \boldsymbol{\alpha} \in \mathbb{Z}_+^k\}. \tag{16}$$

### 3.3 Defining the Sobolev Spaces

For given $\zeta, m \in \mathbb{Z}_+$, let $\mathcal{C}^{\zeta,m}((-1,1), \mathbb{R}^k; \mathbb{R}^{p \times p})$ be the space of functions $\boldsymbol{R}$ defined in $(-1, 1) \times \mathbb{R}^k$, with values in $\mathbb{R}^{p \times p}$, such that $\frac{\mathrm{d}^j}{\mathrm{d}s^j} \partial^{\boldsymbol{\beta}} \boldsymbol{R}$ exist and are continuous for $j = 0, 1, \ldots, \zeta$ and $0 \leq |\boldsymbol{\beta}| \leq m$, where $\frac{\mathrm{d}}{\mathrm{d}s}$ represents the differentiation in $(-1, 1)$ and $\partial^{\boldsymbol{\beta}}$ the partial derivative in $\mathbb{R}^k$ of multi-index $\boldsymbol{\beta}$.

Define

$$\|\boldsymbol{R}\|_{W_{d,k}^{\zeta,m}}^2 := \sum_{j=0}^{\zeta} \sum_{|\boldsymbol{\beta}| \leq m} \int_{\mathbb{R}^k} \int_{-1}^1 \left\| \frac{\mathrm{d}^j}{\mathrm{d}s^j} \partial^{\boldsymbol{\beta}} \boldsymbol{R}(s, \boldsymbol{h}) \right\|_F^2 \left(1 - s^2\right)^{d/2-1+j} \, \mathrm{d}s \, \boldsymbol{\nu}(\mathrm{d}\boldsymbol{h}), \tag{17}$$

where $\|\cdot\|_F$ is the Fröebenius norm in $\mathbb{R}^{p \times p}$, induced by the Fröebenius inner product $\langle A, B \rangle_F := \mathrm{tr}(AB^\top)$, so that $\|A\|_F^2 = \mathrm{tr}(AA^\top)$, with tr denoting the trace operator.

Finally, define the Sobolev space $W_{d,k}^{\zeta,m}((-1,1), \mathbb{R}^k; \mathbb{R}^{p \times p})$ as the completion of the space $\mathcal{C}^{\zeta,m}((-1,1), \mathbb{R}^k; \mathbb{R}^{p \times p})$ with respect to the norm (17) (with the usual identification of a.e. equal functions).

**Remark 3.1.** *The choice of this particular norm is due to the actual meaning of the variables in our setting: in fact the differentiation with respect to $s$ is connected to a differentiation on the sphere with respect to the geodesic distance, defined as $\arccos(\langle \cdot, \cdot \rangle)$ between any pair of points on the spherical shell. The measure $\left(1 - s^2\right)^{d/2-1} \mathrm{d}s$ corresponds to the surface (Haar) measure $\sigma_d$ on $\mathbb{S}^d$.*

### 3.4 Quantifying Smoothness

In the following we intend to obtain estimates from below and from above for the Sobolev norm (17). We will use the equivalence relation $f \sim g$ to relate functions $f, g$, meaning that $cg \leq f \leq Cg$ with constants $c, C > 0$ that can only depend on $(d, k, \zeta, m)$. Note that this is the case if the constants depend also on $j$ or on $\boldsymbol{\beta}$, since it will always be intended that $j \leq \zeta$ and $|\boldsymbol{\beta}| \leq m$, so they only take values in a finite set depending on $\zeta, m, k$.

Our search from smoothness starts by defining a proper spectral inversion of $\boldsymbol{K}$ under the Fröebenius norm $\| \cdot \|_F^2$. To do so, for $\boldsymbol{\beta} \in \mathbb{Z}_+^k$ and $j \in \mathbb{Z}_+$ such that $|\boldsymbol{\beta}| \leq m$ and $j \leq \zeta$, we define

$$I_{j,\boldsymbol{\beta}} := \int_{\mathbb{R}^k} \int_{-1}^1 \left\| \frac{\mathrm{d}^j}{\mathrm{d}s^j} \partial^{\boldsymbol{\beta}} \boldsymbol{K}(s, \boldsymbol{h}) \right\|_F^2 \left(1 - s^2\right)^{d/2-1+j} \, \mathrm{d}s \, \boldsymbol{\nu}(\mathrm{d}\boldsymbol{h}). \tag{18}$$

We now define a sequence $\{s_{j,\boldsymbol{\beta}}\}_{j,\boldsymbol{\beta}}$ with generic element $s_{j,\boldsymbol{\beta}}$ being identically equal to

$$s_{j,\boldsymbol{\beta}} := \sum_{n=j}^{\infty} \sum_{\boldsymbol{\alpha} \geq \boldsymbol{\beta}} \left\| \gamma_{n,\boldsymbol{\alpha}}^d \right\|_F^2 \ (n+1)^{d-1+2j} \boldsymbol{\alpha}^{\boldsymbol{\beta}}. \tag{19}$$

We are going to prove that the quantities (18) and (19) are actually related, and that they are both crucial to quantify smoothness.

We start with a technical result that clearly illustrates the relation between these two quantities.

**Proposition 3.1.** *Let $\zeta, m \in \mathbb{Z}_+$. Given the continuous kernel $\boldsymbol{K} : (-1,1) \times \mathbb{R}^k \to \mathbb{R}^{p \times p}$ that is isotropic on $\mathbb{S}^d$ and stationary on $\mathbb{R}^k$ as in Equation (15), we have that*

$$\|\boldsymbol{K}\|_{W_{d,k}^{\zeta,m}}^2 = \sum_{j=0}^{\zeta} \sum_{|\boldsymbol{\beta}| \leq m} I_{j,\boldsymbol{\beta}} \sim \sum_{j=0}^{\zeta} \sum_{|\boldsymbol{\beta}| \leq m} s_{j,\boldsymbol{\beta}}.$$

*Hence, $\|\boldsymbol{K}\|_{W_{d,k}^{\zeta,m}}^2 < \infty$ if and only if $s_{j,\boldsymbol{\beta}} < \infty$, for all $j \leq \zeta$ and $\boldsymbol{\beta} \in \mathbb{Z}_+^k$ such that $|\boldsymbol{\beta}| \leq m$.*

Proposition 3.1 derives from Lemma A.2 given in Appendix A. Clearly, it does not provide a friendly way to check when a given function $\boldsymbol{K}$ belongs to the Sobolev space $W_{d,k}^{\zeta,m}$ for given quadruple $(d, k, \zeta, m)$ of suitable integers. The next result (see the proof in Appendix A.2) provides an estimate that helps shedding some light in this direction.

**Proposition 3.2.** *In the conditions of Proposition 3.1,*

$$\|\boldsymbol{K}\|_{W_{d,k}^{\zeta,m}}^2 \sim s_{0,\mathbf{0}} + s_{\zeta,\mathbf{0}} + \sum_{|\boldsymbol{\beta}'|=m} s_{0,\boldsymbol{\beta}'} + \sum_{|\boldsymbol{\beta}'|=m} s_{\zeta,\boldsymbol{\beta}'}.$$

A further step ahead can be done by introducing the space of square summable multi-sequences, with respect to a measure $\mu$ in the set $\Omega_k$ defined in Equation (16):

$$\ell^2(\mu) := \left\{ \{\boldsymbol{\gamma}_{n,\boldsymbol{\alpha}}\}_{(n,\boldsymbol{\alpha}) \in \Omega_k} \subset \mathbb{C}^{p \times p} \ : \ \sum_{n=0}^{\infty} \sum_{\boldsymbol{\alpha} \geq \mathbf{0}} \|\boldsymbol{\gamma}_{n,\boldsymbol{\alpha}}\|_F^2 \ \mu_{n,\boldsymbol{\alpha}} < \infty \right\}.$$

We are ready to state the main result (see the proof in Appendix A.2), which completes our quest for smoothness over product spaces, giving a condition on the Gegenbauer-Hermite spectrum $\left\{ \boldsymbol{\gamma}_{n,\boldsymbol{\alpha}}^d \right\}_{(n,\boldsymbol{\alpha}) \in \Omega_k}$ of the $p$-variate kernel $\boldsymbol{K}$, which holds true if and only if $\boldsymbol{K} \in W_{d,k}^{\zeta,m}((-1,1), \mathbb{R}^k; \mathbb{R}^{p \times p})$, thus quantifying its smoothness in terms of the two parameters $\zeta, m \in \mathbb{Z}_+$.

**Theorem 3.3.** *Let $\zeta, m \in \mathbb{Z}_+$ and the measure $\widetilde{\mu}^{\zeta,m}$ be defined as*

$$\widetilde{\mu}_{n,\boldsymbol{\alpha}}^{\zeta,m} = (n+1)^{d-1} \left[ 1 + (n+1)^{2\zeta} \chi_{n \geq \zeta} \right] \left[ 1 + \sum_{|\boldsymbol{\beta}'|=m} \boldsymbol{\alpha}^{\boldsymbol{\beta}'} \chi_{\boldsymbol{\alpha} \geq \boldsymbol{\beta}'} \right], \quad (n, \boldsymbol{\alpha}) \in \Omega_k, \tag{20}$$

*with $\chi_{n \geq \zeta}$ and $\chi_{\boldsymbol{\alpha} \geq \boldsymbol{\beta}'}$ being equal to 1 if $n \geq \zeta$ and $\boldsymbol{\alpha} \geq \boldsymbol{\beta}'$, respectively, and to 0 otherwise. Then, for a given continuous kernel $\boldsymbol{K} : (-1,1) \times \mathbb{R}^k \to \mathbb{R}^{p \times p}$ that is isotropic on $\mathbb{S}^d$ and stationary on $\mathbb{R}^k$ as in Equation (15), we have that $\boldsymbol{K}$ belongs to the space $W_{d,k}^{\zeta,m}$ if and only if $\left\{ \gamma_{n,\boldsymbol{\alpha}}^d \right\} \in \ell^2(\widetilde{\mu}^{\zeta,m})$.*

Hence, we have proved that, under the spectral construction proposed in this paper for a Gaussian measure space, quantifying smoothness is equivalent to prove summability conditions for the matrices $\gamma_{n,\boldsymbol{\alpha}}^d$. One can certainly argue that these conditions are analytically tricky to check. Yet, Theorem 3.3 provides the building block to deduce the simpler condition below (see the proof in Appendix A.2).

**Corollary 3.4.** *Consider $\overline{\mu}^{\zeta,m}$ one of the following measures in $\Omega_k$*

$$\overline{\mu}_{n,\boldsymbol{\alpha}}^{\zeta,m} = (n+1)^{d-1}\left[1+(n+1)^{2\zeta}\right]\left[1+\sum_{|\boldsymbol{\beta}'|=m}\boldsymbol{\alpha}^{\boldsymbol{\beta}'}\right], \quad (n,\boldsymbol{\alpha})\in\Omega_k, \tag{21}$$

*or*

$$\overline{\mu}_{n,\boldsymbol{\alpha}}^{\zeta,m} = (n+1)^{d-1}\left[1+(n+1)^{2\zeta}\right]\left[1+|\boldsymbol{\alpha}|^m\right] \quad (n,\boldsymbol{\alpha})\in\Omega_k. \tag{22}$$

*If $\left\{\gamma_{n,\boldsymbol{\alpha}}^d\right\}\in\ell^2(\overline{\mu}^{\zeta,m})$, then $\boldsymbol{K}$ belongs to the space $W_{d,k}^{\zeta,m}$.*

**Remark 3.2.** *Our work is general, and the following special cases are covered:*

1. *The case $k=0$ and $p=1$ has been considered by Lang and Schwab (2013).*

2. *The case $k=1$ and $p=1$ has been considered by Clarke De la Cerda et al. (2018).*

3. *The case $d=0$, general case, but the domain restricted to $\mathbb{B}_k\subset\mathbb{R}^k$, with $\mathbb{B}_k$ denoting the $k$-dimensional ball, has been considered by Cleanthous (2023).*

*The work of Cleanthous (2023) is the first, to our knowledge, where multivariate smoothness is considered in the literature.*

## 4 Example

For $d>1$, $a\geq 0$, $b>0$ and $\eta\in(0,1)$, consider the following univariate nonseparable kernel (Emery et al., 2021):

$$\boldsymbol{K}(s,\boldsymbol{h};a,b,\eta) = \frac{(1-\eta)^{d-1}\exp(-b\|\boldsymbol{h}\|^2)}{(1-2\eta s\exp(-a\|\boldsymbol{h}\|^2)+\eta^2\exp(-2a\|\boldsymbol{h}\|^2)^{(d-1)/2}}, \quad s\in[-1,1], \quad \boldsymbol{h}\in\mathbb{R}^k.$$

To calculate its Gegenbauer-Hermite spectrum, we start with the Gegenbauer expansion (Emery et al., 2021)

$$\boldsymbol{K}(s,\boldsymbol{h};a,b,\eta) = (1-\eta)^{d-1}\sum_{n=0}^{\infty}\eta^n\exp(-(an+b)\|\boldsymbol{h}\|^2)G_n^{(d-1)/2}(s)$$

$$= \sum_{n=0}^{\infty}\dim(\mathcal{H}_n^d)C_n^d(\boldsymbol{h};a,b,\eta)\mathcal{G}_n^{(d-1)/2}(s), \quad s\in[-1,1], \quad \boldsymbol{h}\in\mathbb{R}^k,$$

with

$$C_n^d(\boldsymbol{h};a,b,\eta) = \frac{(d-1)(1-\eta)^{d-1}\eta^n}{2n+d-1}\exp(-(an+b)\|\boldsymbol{h}\|^2).$$

The Gegenbauer-Hermite spectrum is given by Equation (14). Accounting for the properties of Hermite polynomials (Magnus et al., 1966, Section 5.6.2), one finds

$$\gamma_{n,\boldsymbol{\alpha}}^d = \int_{\mathbb{R}^k}C_n^d(\boldsymbol{h};a,b,\eta)\Phi_{\boldsymbol{\alpha}}(\boldsymbol{h})\mathrm{d}\boldsymbol{h}$$

$$= \begin{cases} 0 \text{ if one or more components of } \boldsymbol{\alpha} \text{ is odd} \\ \frac{(d-1)(1-\eta)^{d-1}\eta^n}{2n+d-1}\frac{2^{-|\boldsymbol{\alpha}|/2}}{(\boldsymbol{\alpha}/2)!}\sqrt{\frac{\pi^k\boldsymbol{\alpha}!}{(1/2+an+b)^k}}\left(\frac{1}{1+2(an+b)}-1\right)^{|\boldsymbol{\alpha}|/2} \text{ otherwise.} \end{cases}$$

Using the duplication formula for the gamma function (Olver et al., 2010, formula 5.5.5), it is seen that $\frac{2^{-|\boldsymbol{\alpha}|/2}}{(\boldsymbol{\alpha}/2)!}\sqrt{\frac{\pi^k\boldsymbol{\alpha}!}{(1/2+an+b)^k}}$ belongs to $(0,(2\pi)^{k/2}]$. Since, furthermore, $(\frac{1}{1+2(an+b)}-1)$ and $\eta$ belong to $(-1,1)$ and $(0,1)$, respectively, it follows that $\left\{\gamma_{n,\boldsymbol{\alpha}}^d\right\}\in\ell^2(\widetilde{\mu}^{\zeta,m})$ for any $\zeta,m\in\mathbb{Z}_+$. Accordingly, owing to Theorem 3.3, $\boldsymbol{K}$ belongs to $W_{d,k}^{\zeta,m}$ for any $\zeta,m\in\mathbb{Z}_+$.

## 5    Conclusions

Our work provides the foundations to smoothness quantification of Gaussian processes defined over some specific product space involving a $d$-dimensional sphere. Some comments are in order. The results presented in Section 3 can be extended to product spaces involving other manifolds. For instance, classic harmonic analysis arguments prove that the $d$-dimensional sphere might be replaced by a compact two-point homogeneous space at the expense of replacing the normalized Gegenbauer polynomials in Equation (3) with their counterpart over such spaces, known as Jacobi polynomials (Cleanthous et al., 2020). We are not aware of whether our results would hold for other general networks such as graphs with Euclidean edges (Porcu et al., 2023). For such cases, even spectral representations become questionable, so that more mathematical effort is needed in this direction.

Future works may involve the verification of the results presented in this paper for specific classes of scalar and matrix-valued kernels, such as the ones proposed by Porcu et al. (2016; 2018), Alegría et al. (2019) and Emery et al. (2021).

Also, extensions to our work to kernels that are not isotropic on the sphere could be based on spectral characterizations such as the one proposed by Jones (1963) for axially symmetric processes on $\mathbb{S}^2$, i.e., processes that are stationary over longitudes, but not over latitudes, of the 2-sphere. Having some insight in this direction would help to overcome the restrictive assumption of isotropy and allow for wider classes of kernels in vector Gaussian process regression.

## Acknowledgments

This work was partially funded by the São Paulo Research Foundation (FAPESP, Brazil), through grant 2021/04269-0 (APP) and grant 2022/16407-1 (EM), and by the National Agency for Research and Development of Chile, through grants ANID PIA AFB230001 and ANID Fondecyt 1210050 (XE). The authors are very grateful the Action Editors and the Reviewers for their thorough reviews that have allowed for a considerably improved version of the manuscript.

## A  Appendix

### A.1  Technical Lemmas

**Lemma A.1.** *Let $\boldsymbol{\alpha}, \boldsymbol{\beta}, \boldsymbol{\varepsilon} \in \mathbb{Z}_+^k$. If $\boldsymbol{\alpha}+\boldsymbol{\varepsilon} \geq \boldsymbol{\beta}$, then one has:*

$$\frac{(\boldsymbol{\alpha}+\boldsymbol{\varepsilon})!}{(\boldsymbol{\alpha}+\boldsymbol{\varepsilon}-\boldsymbol{\beta})!} \sim \boldsymbol{\alpha}^{\boldsymbol{\beta}}. \tag{23}$$

*Proof.* The claim holds because $c_1(\beta_i, \varepsilon_i)\alpha_i^{\beta_i} \leq \frac{(\alpha_i+\varepsilon_i)!}{(\alpha_i+\varepsilon_i-\beta_i)!} \leq c_2(\beta_i, \varepsilon_i)\alpha_i^{\beta_i}$, $i = 1, 2, \ldots, k$ (Olver et al., 2010, formula 5.11.12). In particular, the constants depend on $\boldsymbol{\beta}$ and $\boldsymbol{\varepsilon}$. $\qquad\square$

For the next lemma we will need to state some formulas. From Olver et al. (2010, formulas 18.9.19, 18.9.21 and 18.7.4), we have

$$\frac{\mathrm{d}^j}{\mathrm{d}s^j}G_n^\lambda(s) = 2^j \left(\lambda\right)_j G_{n-j}^{\lambda+j}(s) \sim G_{n-j}^{\lambda+j}(s), \quad \forall n \geq j, \quad \forall \lambda > 0, \tag{24}$$

$$\frac{\mathrm{d}}{\mathrm{d}s}\mathcal{G}_n^0(s) = \frac{\mathrm{d}}{\mathrm{d}s}T_n(s) = nG_{n-1}^1(s), \quad \forall n \geq 1, \tag{25}$$

where $(\lambda)_j = \frac{\Gamma(\lambda+j)}{\Gamma(\lambda)}$ is the Pochhammer symbol (Olver et al., 2010, formula 5.2.5). Using Olver et al. (2010, Table 18.3.1 and formula 18.14.4), we get

$$\int_{-1}^1 G_n^\lambda(s)G_{n'}^\lambda(s)\left(1-s^2\right)^{\lambda-1/2} \, \mathrm{d}s = \frac{\pi 2^{1-2\lambda}\Gamma(n+2\lambda)}{n!(n+\lambda)\Gamma(\lambda)^2}\delta_{n,n'}, \quad \forall n, n' \geq 0, \quad \forall \lambda > 0, \tag{26}$$

$$\int_{-1}^1 \mathcal{G}_n^0(s)\mathcal{G}_{n'}^0(s)\left(1-s^2\right)^{-1/2} \, \mathrm{d}s \sim \delta_{n,n'}, \quad \forall n, n' \geq 0. \tag{27}$$

Finally, from Muller (1966, equation 11),

$$\frac{\dim(\mathcal{H}_n^d)}{G_n^{(d-1)/2}(1)} = \frac{2n+d-1}{d-1}, \quad \forall n \geq 1, \quad \forall d > 1, \tag{28}$$

$$\dim(\mathcal{H}_n^1) = 2, \quad \forall n \geq 1.$$

**Lemma A.2.** *Let $\zeta, m \in \mathbb{Z}_+$. For $\boldsymbol{\beta} \in \mathbb{Z}_+^k$ and $j \in \mathbb{Z}_+$ such that $|\boldsymbol{\beta}| \leq m$ and $j \leq \zeta$, define $I_{j,\boldsymbol{\beta}}$ and $s_{j,\boldsymbol{\beta}}$ as per Equations (18) and (19). Then the following estimates hold:*

$$I_{j,\boldsymbol{\beta}} \sim \sum_{n=j}^\infty \sum_{\boldsymbol{\alpha} \geq \boldsymbol{\beta}} \left\|\gamma_{n,\boldsymbol{\alpha}}^d\right\|_F^2 (n+1)^{d-1+2j}\frac{\boldsymbol{\alpha}!}{(\boldsymbol{\alpha}-\boldsymbol{\beta})!} \tag{29}$$

*and*

$$I_{j,\boldsymbol{\beta}} \sim s_{j,\boldsymbol{\beta}}. \tag{30}$$

*Proof.* By Equation (15),

$$
\begin{aligned}
I_{j,\boldsymbol{\beta}} &= \int_{\mathbb{R}^k} \int_{-1}^1 \left\|\sum_{n=0}^\infty \dim(\mathcal{H}_n^d) \sum_{|\boldsymbol{\alpha}|\geq 0} \gamma_{n,\boldsymbol{\alpha}}^d \, \partial^{\boldsymbol{\beta}}\Phi_{\boldsymbol{\alpha}}(\boldsymbol{h})\frac{\mathrm{d}^j}{\mathrm{d}s^j}\mathcal{G}_n^{(d-1)/2}(s)\right\|_F^2 \left(1-s^2\right)^{d/2-1+j} \, \mathrm{d}s \, \boldsymbol{\nu}(\mathrm{d}\boldsymbol{h}) \\
&= \sum_{n,n'=0}^\infty \sum_{|\boldsymbol{\alpha}|,|\boldsymbol{\alpha}'|\geq 0} \langle\gamma_{n,\boldsymbol{\alpha}}^d \, \gamma_{n',\boldsymbol{\alpha}'}^d\rangle_F \, \widetilde{J}_{n,n'} J_{\boldsymbol{\alpha},\boldsymbol{\alpha}'},
\end{aligned}
$$

where

$$\widetilde{J}_{n,n'} := \dim(\mathcal{H}_n^d)\dim\left(\mathcal{H}_{n'}^d\right) \int_{-1}^1 \frac{\mathrm{d}^j}{\mathrm{d}s^j}\mathcal{G}_n^{(d-1)/2}(s)\frac{\mathrm{d}^j}{\mathrm{d}s^j}\mathcal{G}_{n'}^{(d-1)/2}(s)\left(1-s^2\right)^{d/2-1+j}\ \mathrm{d}s$$

and

$$J_{\boldsymbol{\alpha},\boldsymbol{\alpha}'} := \int_{\mathbb{R}^k} \partial^{\boldsymbol{\beta}}\Phi_{\boldsymbol{\alpha}}(\boldsymbol{h})\partial^{\boldsymbol{\beta}}\Phi_{\boldsymbol{\alpha}'}(\boldsymbol{h})\ \boldsymbol{\nu}(\mathrm{d}\boldsymbol{h}).$$

Note that, $\widetilde{J}_{n,n'} = 0$ for $n < j$. For $n \geq j \geq 0$, we distinguish two cases, depending on whether $d$ is greater than 1 or not. First, let us examine the case when $d > 1$ and $n \geq j \geq 0$. In this case, we have, by Equation (24),

$$\begin{aligned}
\widetilde{J}_{n,n'} &= \frac{\dim(\mathcal{H}_n^d)}{G_n^{(d-1)/2}(1)}\frac{\dim\left(\mathcal{H}_{n'}^d\right)}{G_{n'}^{(d-1)/2}(1)} \int_{-1}^1 \frac{\mathrm{d}^j}{\mathrm{d}s^j}G_n^{(d-1)/2}(s)\frac{\mathrm{d}^j}{\mathrm{d}s^j}G_{n'}^{(d-1)/2}(s)\left(1-s^2\right)^{d/2-1+j}\ \mathrm{d}s \\
&\sim \frac{\dim(\mathcal{H}_n^d)}{G_n^{(d-1)/2}(1)}\frac{\dim\left(\mathcal{H}_{n'}^d\right)}{G_{n'}^{(d-1)/2}(1)} \int_{-1}^1 G_{n-j}^{(d-1)/2+j}(s)G_{n'-j}^{(d-1)/2+j}(s)\left(1-s^2\right)^{d/2-1+j}\ \mathrm{d}s.
\end{aligned}$$

By Equations (26) and (28), we obtain

$$\begin{aligned}
\widetilde{J}_{n,n'} &\sim \left(\frac{2n+d-1}{d-1}\right)\left(\frac{2n'+d-1}{d-1}\right)\int_{-1}^1 G_{n-j}^{(d-1)/2+j}(s)G_{n'-j}^{(d-1)/2+j}(s)\left(1-s^2\right)^{d/2-1+j}\ \mathrm{d}s \\
&= \left(\frac{2n+d-1}{d-1}\right)^2 \frac{\pi 2^{2-d-2j}\Gamma(n+j+d-1)}{(n-j)!(n+\frac{d-1}{2})\Gamma(\frac{d-1}{2}+j)^2}\delta_{n,n'}.
\end{aligned}$$

Since

$$\frac{2n+d-1}{d-1} \sim n+1$$

and (Olver et al., 2010, formula 5.11.12)

$$\frac{\pi 2^{2-d-2j}\Gamma(n+j+d-1)}{(n-j)!(n+\frac{d-1}{2})\Gamma(\frac{d-1}{2}+j)^2} \sim (n+1)^{d-3+2j},$$

the previous result simplifies into

$$\widetilde{J}_{n,n'} \sim (n+1)^{d-1+2j}\delta_{n,n'}. \tag{31}$$

Let us now address the case when $d = 1$. For $n \geq j = 0$, we have

$$\begin{aligned}
\widetilde{J}_{n,n'} &= \dim(\mathcal{H}_n^1)\dim\left(\mathcal{H}_{n'}^1\right)\int_{-1}^1 \mathcal{G}_n^0(s)\mathcal{G}_{n'}^0(s)\left(1-s^2\right)^{-1/2}\ \mathrm{d}s \\
&\sim \delta_{n,n'},
\end{aligned}$$

based on Equation (27). For $n \geq j > 0$, we have, by Equations (23), (24), (25) and (26):

$$\begin{aligned}
\widetilde{J}_{n,n'} &= \dim(\mathcal{H}_n^1)\dim\left(\mathcal{H}_{n'}^1\right)\int_{-1}^1 \frac{\mathrm{d}^j}{\mathrm{d}s^j}\mathcal{G}_n^0(s)\frac{\mathrm{d}^j}{\mathrm{d}s^j}\mathcal{G}_{n'}^0(s)\left(1-s^2\right)^{j-1/2}\ \mathrm{d}s \\
&= \dim(\mathcal{H}_n^1)\dim\left(\mathcal{H}_{n'}^1\right)nn'4^{j-1}[(j-1)!]^2\int_{-1}^1 G_{n-j}^j(s)G_{n'-j}^j(s)\left(1-s^2\right)^{j-1/2}\ \mathrm{d}s \\
&= \dim(\mathcal{H}_n^1)\dim\left(\mathcal{H}_{n'}^1\right)n\frac{\pi(n+j-1)!}{2(n-j)!}\delta_{n,n'} \\
&\sim (n+1)^{2j}\delta_{n,n'}.
\end{aligned}$$

Hence, Equation (31) remains valid when $d = 1$.

On the other hand,

$$
\begin{aligned}
J_{\boldsymbol{\alpha},\boldsymbol{\alpha}'} &= \int_{\mathbb{R}^k} \partial^{\boldsymbol{\beta}} \prod_{j=1}^{k} H_{\alpha_j}(h_j) \, \partial^{\boldsymbol{\beta}} \prod_{j=1}^{k} H_{\alpha'_j}(h_j) \; \boldsymbol{\nu}(\mathrm{d}\boldsymbol{h}) \\
&= \int_{\mathbb{R}^k} \prod_{j=1}^{k} \partial^{\beta_j} H_{\alpha_j}(h_j) \prod_{j=1}^{k} \partial^{\beta_j} H_{\alpha'_j}(h_j) \; \boldsymbol{\nu}(\mathrm{d}\boldsymbol{h}) \\
&= \int_{\mathbb{R}^k} \prod_{j=1}^{k} \sqrt{\frac{\alpha_j!}{(\alpha_j - \beta_j)!}}\, H_{\alpha_j - \beta_j}(h_j) \prod_{j=1}^{k} \sqrt{\frac{\alpha'_j!}{(\alpha'_j - \beta_j)!}}\, H_{\alpha'_j - \beta_j}(h_j) \; \boldsymbol{\nu}(\mathrm{d}\boldsymbol{h}) \\
&= \sqrt{\frac{\boldsymbol{\alpha}!}{(\boldsymbol{\alpha}-\boldsymbol{\beta})!}} \sqrt{\frac{\boldsymbol{\alpha}'!}{(\boldsymbol{\alpha}'-\boldsymbol{\beta})!}} \int_{\mathbb{R}^k} \prod_{j=1}^{k} H_{\alpha_j - \beta_j}(h_j) \prod_{j=1}^{k} H_{\alpha'_j - \beta_j}(h_j) \; \boldsymbol{\nu}(\mathrm{d}\boldsymbol{h}) \\
&= \sqrt{\frac{\boldsymbol{\alpha}!}{(\boldsymbol{\alpha}-\boldsymbol{\beta})!}} \sqrt{\frac{\boldsymbol{\alpha}'!}{(\boldsymbol{\alpha}'-\boldsymbol{\beta})!}} \int_{\mathbb{R}^k} \Phi_{\boldsymbol{\alpha}-\boldsymbol{\beta}}(\boldsymbol{h}) \, \Phi_{\boldsymbol{\alpha}'-\boldsymbol{\beta}}(\boldsymbol{h}) \; \boldsymbol{\nu}(\mathrm{d}\boldsymbol{h})
\end{aligned}
$$

and then, since the multivariate Hermite polynomials are an orthonormal basis of $L^2(\mathbb{R}^k, \boldsymbol{\nu})$,

$$
J_{\boldsymbol{\alpha},\boldsymbol{\alpha}'} = \frac{\boldsymbol{\alpha}!}{(\boldsymbol{\alpha}-\boldsymbol{\beta})!} \delta_{\boldsymbol{\alpha},\boldsymbol{\alpha}'}, \quad \boldsymbol{\alpha} \geq \boldsymbol{\beta}. \tag{32}
$$

Thus, from Equations (31) and (32) we obtain Equation (29) and then Equation (30) using Equation (23). $\qquad\square$

**Lemma A.3.** *Let* $\boldsymbol{\alpha}, \boldsymbol{\beta}, \boldsymbol{\beta}' \in \mathbb{Z}_+^k$. *If* $\boldsymbol{\alpha} \geq \boldsymbol{\beta}' \geq \boldsymbol{\beta} \geq \mathbf{0}$, *then* $\boldsymbol{\alpha}^{\boldsymbol{\beta}} \leq \boldsymbol{\alpha}^{\boldsymbol{\beta}'}$.

*Proof.* In the scalar case, $a, b, b' \in \mathbb{Z}_+$ with $a \geq b' \geq b$ implies $a^b \leq a^{b'}$. By applying this to each component we obtain the claim. $\qquad\square$

Fix $\boldsymbol{\beta} \in \mathbb{Z}_+$ with $|\boldsymbol{\beta}| \leq m$, let

$$
I_{\boldsymbol{\beta}} := \{ \boldsymbol{\beta}' \in \mathbb{Z}_+ : \; \boldsymbol{\beta}' \geq \boldsymbol{\beta}, \; |\boldsymbol{\beta}'| = m \}
$$

and

$$
A_{\boldsymbol{\beta}} := \{ \boldsymbol{\alpha} \in \mathbb{Z}_+ : \; \boldsymbol{\alpha} \geq \boldsymbol{\beta} \}.
$$

**Lemma A.4.** *The set* $A_{\boldsymbol{\beta}}$ *can be written as*

$$
A_{\boldsymbol{\beta}} = \widetilde{A}_{\boldsymbol{\beta}} \cup \bigcup_{\boldsymbol{\beta}' \in I_{\boldsymbol{\beta}}} A_{\boldsymbol{\beta}'} \tag{33}
$$

*where* $\widetilde{A}_{\boldsymbol{\beta}} = \{ \boldsymbol{\alpha} \in \mathbb{Z}_+ : |\boldsymbol{\alpha}| < m, \; \boldsymbol{\alpha} \geq \boldsymbol{\beta} \}$.

*Proof.* If $\boldsymbol{\alpha} \in A_{\boldsymbol{\beta}}$, then either $|\boldsymbol{\alpha}| < m$ or there exists $\boldsymbol{\beta}' \in I_{\boldsymbol{\beta}}$ such that $\boldsymbol{\beta} \leq \boldsymbol{\beta}' \leq \boldsymbol{\alpha}$. One can construct such $\boldsymbol{\beta}'$ by increasing those components $\beta_i$ of $\boldsymbol{\beta}$ that satisfy $\beta_i < \alpha_i$ until reaching $|\boldsymbol{\beta}'| = m$. $\qquad\square$

## A.2 Proofs of the main statements

*Proof of Proposition 3.2.* Given $j \leq \zeta$ and $|\boldsymbol{\beta}| \leq m$, in the definition (19) of $s_{j,\boldsymbol{\beta}}$, the sum runs over every $n \geq j$ and $\alpha \in A_{\boldsymbol{\beta}}$.
We have that

- if $n \geq \zeta$ then $(n+1)^{2j} \leq (n+1)^{2\zeta}$,

- if $n < \zeta$ then $(n+1)^{2j} \leq (\zeta+1)^{2\zeta}$.

Moreover, by Equation (33), if $\boldsymbol{\alpha} \in A_{\boldsymbol{\beta}}$ then either $\boldsymbol{\alpha} \in \widetilde{A}_{\boldsymbol{\beta}}$ or $\boldsymbol{\alpha} \in A_{\boldsymbol{\beta}'}$ for some $\boldsymbol{\beta}' \in I_{\boldsymbol{\beta}}$:

- if $\boldsymbol{\alpha} \in \widetilde{A}_{\boldsymbol{\beta}}$ then $\boldsymbol{\alpha}^{\boldsymbol{\beta}} \leq m^m$,

- if $\boldsymbol{\alpha} \in A_{\boldsymbol{\beta}'}$ with $\boldsymbol{\beta}' \in I_{\boldsymbol{\beta}}$, then by Lemma A.3, it holds $\boldsymbol{\alpha}^{\boldsymbol{\beta}} \leq \boldsymbol{\alpha}^{\boldsymbol{\beta}'}$.

Then we can estimate from above all the terms $\left\| \boldsymbol{\gamma}_{n,\boldsymbol{\alpha}}^d \right\|_F^2 (n+1)^{d-1+2j} \boldsymbol{\alpha}^{\boldsymbol{\beta}}$ in the definition (19) of $s_{j,\boldsymbol{\beta}}$ with the corresponding term

- in $s_{\zeta,\boldsymbol{\beta}'}$ with $\boldsymbol{\alpha} \geq \boldsymbol{\beta}' \geq \boldsymbol{\beta}$ if $|\boldsymbol{\alpha}| \geq m$, $n \geq \zeta$,

- in $(\zeta+1)^{2\zeta} s_{0,\boldsymbol{\beta}'}$ with $\boldsymbol{\alpha} \geq \boldsymbol{\beta}' \geq \boldsymbol{\beta}$ if $|\boldsymbol{\alpha}| \geq m$, $n < \zeta$,

- in $m^m s_{\zeta,\boldsymbol{0}}$ if $|\boldsymbol{\alpha}| < m$, $n \geq \zeta$,

- in $(\zeta+1)^{2\zeta} m^m s_{0,\boldsymbol{0}}$ if $|\boldsymbol{\alpha}| < m$, $n < \zeta$.

As a consequence

$$s_{j,\boldsymbol{\beta}} \leq (\zeta+1)^{2\zeta} m^m s_{0,\boldsymbol{0}} + m^m s_{\zeta,\boldsymbol{0}} + (\zeta+1)^{2\zeta} \sum_{|\boldsymbol{\beta}'|=m} s_{0,\boldsymbol{\beta}'} + \sum_{|\boldsymbol{\beta}'|=m} s_{\zeta,\boldsymbol{\beta}'},$$

and then summing up all the terms (the number of such terms only depends on $\zeta, k, m$), we get

$$\left\| \boldsymbol{K} \right\|_{W_{d,k}^{\zeta,m}}^2 \sim \sum_{j=0}^{\zeta} \sum_{|\boldsymbol{\beta}|\leq m} s_{j,\boldsymbol{\beta}} \sim s_{0,\boldsymbol{0}} + s_{\zeta,\boldsymbol{0}} + \sum_{|\boldsymbol{\beta}'|=m} s_{0,\boldsymbol{\beta}'} + \sum_{|\boldsymbol{\beta}'|=m} s_{\zeta,\boldsymbol{\beta}'},$$

since the estimate from below is trivial. □

*Proof of Theorem 3.3.* By Proposition 3.2, all we have to do is to prove that

$$s_{0,\boldsymbol{0}} + s_{\zeta,\boldsymbol{0}} + \sum_{|\boldsymbol{\beta}'|=m} s_{0,\boldsymbol{\beta}'} + \sum_{|\boldsymbol{\beta}'|=m} s_{\zeta,\boldsymbol{\beta}'} = \sum_{n=0}^{\infty} \sum_{\boldsymbol{\alpha} \geq \boldsymbol{0}} \left\| \boldsymbol{\gamma}_{n,\boldsymbol{\alpha}}^d \right\|_F^2 \widetilde{\mu}_{n,\boldsymbol{\alpha}}^{\zeta,m}, \tag{34}$$

where $\widetilde{\mu}_{n,\boldsymbol{\alpha}}^{\zeta,m}$ is given in Equation (20). Indeed, from Equation (19), one has:

$$s_{0,\boldsymbol{0}} = \sum_{n=0}^{\infty} \sum_{\boldsymbol{\alpha} \geq \boldsymbol{0}} \left\| \boldsymbol{\gamma}_{n,\boldsymbol{\alpha}}^d \right\|_F^2 (n+1)^{d-1} \boldsymbol{\alpha}^{\boldsymbol{0}},$$

$$s_{\zeta,\boldsymbol{\beta}'} = \sum_{n=\zeta}^{\infty} \sum_{\boldsymbol{\alpha} \geq \boldsymbol{\beta}'} \left\| \boldsymbol{\gamma}_{n,\boldsymbol{\alpha}}^d \right\|_F^2 (n+1)^{d-1+2\zeta} \boldsymbol{\alpha}^{\boldsymbol{\beta}'} = \sum_{n=0}^{\infty} \sum_{\boldsymbol{\alpha} \geq \boldsymbol{0}} \left\| \boldsymbol{\gamma}_{n,\boldsymbol{\alpha}}^d \right\|_F^2 (n+1)^{d-1+2\zeta} \chi_{n \geq \zeta} \, \boldsymbol{\alpha}^{\boldsymbol{\beta}'} \chi_{\boldsymbol{\alpha} \geq \boldsymbol{\beta}'},$$

$$s_{\zeta,\boldsymbol{0}} = \sum_{n=\zeta}^{\infty} \sum_{\boldsymbol{\alpha} \geq \boldsymbol{0}} \left\| \boldsymbol{\gamma}_{n,\boldsymbol{\alpha}}^d \right\|_F^2 (n+1)^{d-1+2\zeta} \boldsymbol{\alpha}^{\boldsymbol{0}} = \sum_{n=0}^{\infty} \sum_{\boldsymbol{\alpha} \geq \boldsymbol{0}} \left\| \boldsymbol{\gamma}_{n,\boldsymbol{\alpha}}^d \right\|_F^2 (n+1)^{d-1+2\zeta} \chi_{n \geq \zeta} \, \boldsymbol{\alpha}^{\boldsymbol{0}},$$

$$s_{0,\boldsymbol{\beta}'} = \sum_{n=0}^{\infty} \sum_{\boldsymbol{\alpha} \geq \boldsymbol{\beta}'} \left\| \boldsymbol{\gamma}_{n,\boldsymbol{\alpha}}^d \right\|_F^2 (n+1)^{d-1} \boldsymbol{\alpha}^{\boldsymbol{\beta}'} = \sum_{n=0}^{\infty} \sum_{\boldsymbol{\alpha} \geq \boldsymbol{0}} \left\| \boldsymbol{\gamma}_{n,\boldsymbol{\alpha}}^d \right\|_F^2 (n+1)^{d-1} \boldsymbol{\alpha}^{\boldsymbol{\beta}'} \chi_{\boldsymbol{\alpha} \geq \boldsymbol{\beta}'},$$

where $\boldsymbol{\alpha}^{\boldsymbol{0}} = 1$. This all adds up into

$$\widetilde{\mu}_{n,\boldsymbol{\alpha}}^{\zeta,m} = (n+1)^{d-1} \left[ 1 + (n+1)^{2\zeta} \chi_{n \geq \zeta} \sum_{|\boldsymbol{\beta}'|=m} \boldsymbol{\alpha}^{\boldsymbol{\beta}'} \chi_{\boldsymbol{\alpha} \geq \boldsymbol{\beta}'} + (n+1)^{2\zeta} \chi_{n \geq \zeta} + \sum_{|\boldsymbol{\beta}'|=m} \boldsymbol{\alpha}^{\boldsymbol{\beta}'} \chi_{\boldsymbol{\alpha} \geq \boldsymbol{\beta}'} \right]$$

$$= (n+1)^{d-1} \left[ 1 + (n+1)^{2\zeta} \chi_{n \geq \zeta} \right] \left[ 1 + \sum_{|\boldsymbol{\beta}'|=m} \boldsymbol{\alpha}^{\boldsymbol{\beta}'} \chi_{\boldsymbol{\alpha} \geq \boldsymbol{\beta}'} \right]. \qquad □$$

*Proof of Corollary 3.4.* Obviously,

$$\widetilde{\mu}_{n,\boldsymbol{\alpha}}^{\zeta,m} \le (n+1)^{d-1} \left[1 + (n+1)^{2\zeta}\right] \left[1 + \sum_{|\boldsymbol{\beta}'|=m} \boldsymbol{\alpha}^{\boldsymbol{\beta}'}\right].$$

Moreover, $\boldsymbol{\alpha}^{\boldsymbol{\beta}'} \le |\boldsymbol{\alpha}|^m$, so that $\sum_{|\boldsymbol{\beta}'|=m} \boldsymbol{\alpha}^{\boldsymbol{\beta}'} \le D(m,k)|\boldsymbol{\alpha}|^m$, where $D(m,k) > 1$ is the number of multi-indices in $\mathbb{Z}_+^k$ of module $m$ (an integer depending only $m$ and $k$). Accordingly,

$$\begin{aligned}
\widetilde{\mu}_{n,\boldsymbol{\alpha}}^{\zeta,m} &\le (n+1)^{d-1} \left[1 + (n+1)^{2\zeta}\right] \left[1 + D(m,k)|\boldsymbol{\alpha}|^m\right] \\
&\le D(m,k)(n+1)^{d-1} \left[1 + (n+1)^{2\zeta}\right] \left[1 + |\boldsymbol{\alpha}|^m\right].
\end{aligned}$$

Thus, considering the measure in Equation (22) or in Equation (21), if $\left\{\boldsymbol{\gamma}_{n,\boldsymbol{\alpha}}^d\right\} \in \ell^2(\overline{\mu}^{\zeta,m})$, then $\left\{\boldsymbol{\gamma}_{n,\boldsymbol{\alpha}}^d\right\} \in \ell^2(\widetilde{\mu}^{\zeta,m})$ and the result follows by Theorem 3.3. $\qquad\square$

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
