# 1 3DrE

Strenghts:

• The presentation is relatively clear, with the "route to smoothness" helping the readers understand the proof process, despite the technicalities.

• While I haven't been able to check every minute detail of the proof, the logic is presented clearly and the arguments are mathematically sound.

Weakness:

• The space that is considered in this paper is oddly specific. The authors simply list a bunch of works where the particular product space is used but making this more explicit (e.g. giving an example in one of these works cited) would help us understand better the value of the paper.

• The work feels incomplete. If I understand correctly, the authors are interested in characterising the smoothness of the Gaussian sample paths but the proof stops at proving the smoothness of the kernel. Some comments or extra arguments on how to go from the kernel regularity to the sample path regularity is necessary in my opinion.

Requested Changes:

I think the work doesn't fully resolve what it sets out to do, which is to characterise the smoothness of Gaussian processes (in my understanding, this means sample paths). Some extra arguments to go from the kernel smoothness to sample path smoothness is necessary.

We are indebted to the Referees for spotting this subtle inconsistency and allowing us to clarify the following.

For the remainder, we call a vector-valued Gaussian random field, $\boldsymbol{Z}$, as non trivially cross correlated when the cross correlations in the covariance matrix-valued function, $\boldsymbol{K}$ are not zero. We note that, under Gaussianity, stochastic independendence is in one-to-one correspondence with zero correlation. For a Gaussian vector-valued random field with independent components, quantifying the smoothness of $\boldsymbol{Z}$ is equivalent to separately quantify the smoothness of each scalar component, $Z_i$, for $i = 1, \ldots, p$. For such a case, the present work is clearly redundant. We challenge the case of nonzero cross-correlations here, and we note that, to the best of our knowledge, a general characterization of smoothness for vector-valued Gaussian fields is not available in the literature. In fact, our Proposition 3.2 shows that all the elements $K_{ij}$ of the covariance $\boldsymbol{K}$ *mix up* to determine the smoothness of the underlying random field. This fact is consistent in the literature, and up to now there is no literature allowing to translate the smoothness of $\boldsymbol{K}$ into the smoothness of $\boldsymbol{Z}$, whatever the space, and whatever the metric. We have eluded this terminology inconsistency by talking about kernel smoothness in the new version of the paper. We hope this fixes things.
We also note the generality of our result: for $p = 1$ and $k = 1$, Sobolev smoothness for a scalar-valued Gaussian random field was covered by Clarke De la Cerda et al. (2018). The case $p = 1$ and $k = 0$ (a scalar Gaussian field on $d$-dimensional spheres only), Sobolev smoothness was characterized by Lang and Schwab (2013). Recently, Cleanthous (2023) considers a different setting of vector-valued Gaussian fields, but the index set is the $d$-dimensional ball.

# 2 v83p

Strengths: the paper is primarily theoretical and the theoretical results are, to the best of my knowledge, novel and interesting.

Weaknesses:

1. The motivation on the study of smoothness properties, and the specific choice of product is a bit vague and unclear; e.g., more connection to applications may be helpful.

2. The paper is rather technical and the presentation can be improved. See my detailed comments below.

Requested Changes:

1. Why the setting of product space between unit sphere and standard Euclidean space? The authors mentioned on possible applications at the end of Section 1.2, but the connections are a bit vague and unclear. In fact, it could be to good idea to showcase the application of the proposed characterization on a concrete ML example.

This point has been taken in due consideration and a detailed list of applications is now available in the new version of the manuscript.

2. It would be helpful to point to, when talking about the contribution of this work at the top of P3, the corresponding precise results in the paper. For example, the route is given in P5, the spectral characterization of smoothness that relies on the properties of the matrix-valued kernel is given in xxx, etc.

We included in each item the section that contains the respective subject.

3. Please refer to the precise section of proof in the Appendix.

Done and Thanks.

A few minor issues:

1. Corollary 3.4: "one of the following measureS"?

Done and Thanks.

2. what does $\ell^2$ mean in Proposition 3.3?

It was already defined in the paragraph above Proposition 3.3 (now renamed as Theorem 3.3).

## 3  wYMA

Strengths: The spectral analysis is well-founded and the paper is carefully written.

Weaknesses: The choice of notation, such as using $\|\cdot\|_k$ for the Euclidean norm and $\|\cdot\|_*$ for the Frobenius norm, might confuse readers familiar with the conventional usage of these notations in other contexts. Also, the reliance on external references, such as Erdélyi (1953) and Cramér's theorem, without fully integrating their details into the discussion, detracts from the paper's self-contained nature and accessibility.

Requested Changes:

The notation $\|\cdot\|_k$ is usually used for the $\ell_k$ norm, and $\|\cdot\|_*$ is usually used for the nuclear norm; please change these notations (it would be fine to use $\|\cdot\|$ for both, or $\|\cdot\|_F$ or $\|\cdot\|_{HS}$ for the Frobenius norm).

Done and Thanks.

Please make Section 2 more self-contained by formally stating and explaining the application of formula 11.4.2 from Erdélyi (1953) and Cramér's theorem within the context of your analysis.

Done and Thanks. The section has been entirely revised to clarify the contents and to be self-contained.

Typos:

Section 1.2: change the title from "Why Studying" to "Why Study".

Done and Thanks.

In eq. (5), the last comma should be a period. After eq. (10): "indexes" -> "indices".

Done and Thanks.

## 4    Reviewer TRSJ

Strength: Novel construction to define smoothness of vector-valued Gaussian process.

Weaknesses: The construction depends on some recent references, e.g. Porcu et al. 2021. And therefore it is not introduced in a self-contained way.

We agree with this comment. At pages 4 and 5 in the new version of the manuscript we have reported how to attain the covariance mappings that are used in the paper in concert with their spectral representation. We hope this helps to have a more self-contained exposition.

Requested Changes:

Somewhere around sec 2, the authors should explain how smoothness is quantified for scalar Gaussian processes, so the readers can see the developments are not trivial. Overall, this paper introduces a construction but does not compare it with alternative constructions.

We thank the referee for providing this constructive criticism. We have modified several statements to make the exposition more self contained and understandable.

For the remainder, we call a vector-valued Gaussian random field, $\boldsymbol{Z}$, as non trivially cross correlated when the cross correlations in the covariance matrix-valued function, $\boldsymbol{K}$ are not zero. We note that, under Gaussianity, stochastic independendence is in one-to-one correspondence with zero correlation. For a Gaussian vector-valued random field with independent components, quantifying the smoothness of $\boldsymbol{Z}$ is equivalent to separately quantify the smoothness of each scalar component, $Z_i$, for $i = 1, \ldots, p$. For such a case, the present work is clearly redundant. We challenge the case of nonzero cross-correlations here, and we note that, to the best of our knowledge, a general characterization of smoothness for vector-valued Gaussian fields is not available in the literature. In fact, our Proposition 3.2 shows that all the elements $K_{ij}$ of the covariance $\boldsymbol{K}$ *mix up* to determine the smoothness of the underlying random field. This fact is consistent in the literature, and up to now there is no literature allowing to translate the smoothness of $\boldsymbol{K}$ into the smoothness of $\boldsymbol{Z}$, whatever the space, and whatever the metric. We have eluded this terminology inconsistency by talking about kernel smoothness in the new version of the paper. We hope this fixes things.
We also note the generality of our result: for $p = 1$ and $k = 1$, Sobolev smoothness for a scalar-valued Gaussian random field was covered by Clarke De la Cerda et al. (2018). The case $p = 1$ and $k = 0$ (a scalar Gaussian field on $d$-dimensional spheres only), Sobolev smoothness was characterized by Lang and Schwab (2013). Recently, Cleanthous (2023) considers a different setting of vector-valued Gaussian fields, but the index set is the $d$-dimensional ball.

How significant is the studied product space? Can either the sphere or the Euclidean space be zero dimension ($d = 0$ or $k = 0$), so that the product space includes the sphere/Euclidean space as special cases?

This is an excellent comment and the answer is yes. Specifically:

1. The case $k = 0$, $p = 1$ has been considered by Lang and Schwab (2013);

2. The case $p = 1$ and $k = 1$ has been considered by Clarke De la Cerda et al. (2018);

3. The case $d = 0$, general case, but the domain restricted to $\mathbb{B}_k \subset \mathbb{R}^k$, with $\mathbb{B}_k$ denoting the $k$-dimensional ball, has been considered by Cleanthous (2023).

4. The work of Cleanthous (2023) is the first, to our knowledge, where multivariate smoothness is considered in the literature.

1.4 introduce $\mathbb{Z}$

Done and Thanks.

P4 "cov" is not introduced

Done and Thanks.

3.1 "It can be namely verified that these functions ..." Add more explanations and intuition.

Done and Thanks.

3.3. Explain why $(-1,1) \times \mathbb{R}^k$ is investigated. Why the interval $(-1,1)$ is not closed? Since we are going to define the Sobolev space as the completion with respect to the integral norm (eq. (12)) and identifying a.e. equal functions, making the definition on $(-1,1)$ or on $[-1,1]$ is the same, as they differ by a set of zero measure.

sec 3. In the end, how smoothness is "quantified"? Is it a binary property (whether or not K belongs to the space $W_{d,k}^{\zeta,m}$)?

Here the answer is yes, and is now stated before Theorem 3.3: the smoothness is "quantified" by the couples $\zeta, m$ for which $K$ belongs to $W_{d,k}^{\zeta,m}$