# OpenReview forum: "Understanding Smoothness of Vector Gaussian Processes on Product Spaces"
_TMLR — Accepted by TMLR_

### Review · Reviewer_3DrE · 2024-02-11

**Summary Of Contributions:**

The contribution of this paper is to quantify precisely, the smoothness of vector Matern GPs defined over a product space, with one part consisting a $d$-sphere and the other consisting a $k$-dimensional Euclidean space. The proof follows by constructing a Karhunen-Loeve expansion with respect to a basis on a particular measure space, one part consisting the Haar measure of the sphere and the other part consisting a Gaussian measure on $\mathbb{R}^k$, then bounding the coefficients in such a way that the kernel belongs to a particular Sobolev-space of matrix-valued functions defined over the product space. The authors provide an explicit bound on said coefficients.

**Audience:**

Yes

**Broader Impact Concerns:**

This paper is mathematical in nature and there are no concerns regarding ethical implications.

**Claims And Evidence:**

Yes

**Requested Changes:**

I think the work doesn't fully resolve what it sets out to do, which is to characterise the smoothness of Gaussian processes (in my understanding, this means sample paths). Some extra arguments to go from the kernel smoothness to sample path smoothness is necessary.

**Strengths And Weaknesses:**

Strenghts:
- The presentation is relatively clear, with the "route to smoothness" helping the readers understand the proof process, despite the technicalities.
- While I haven't been able to check every minute detail of the proof, the logic is presented clearly and the arguments are mathematically sound.

Weakness:
- The space that is considered in this paper is oddly specific. The authors simply list a bunch of works where the particular product space is used but making this more explicit (e.g. giving an example in one of these works cited) would help us understand better the value of the paper.
- The work feels incomplete. If I understand correctly, the authors are interested in characterising the smoothness of the Gaussian sample paths but the proof stops at proving the smoothness of the kernel. Some comments or extra arguments on how to go from the kernel regularity to the sample path regularity is necessary in my opinion.

---

> ### Author Response · Authors · 2024-03-27
> **Replies**
>
> We are extremely grateful to the Editors and the Referees for their great job that allowed for a considerably improved version of the manuscript.
> A carefully revised version, in concert with a Supplement that answers all the points by the Referees, are attached to this second submission.
> Thank you, best wishes,
> The Authors

---

### Review · Reviewer_v83p · 2024-03-03

**Summary Of Contributions:**

In this paper, the authors provide characterization of the smooth properties of vector Gaussian processes defined on the product space of as unit sphere manifold and standard Euclidean space.
Through a spectral representation approach (e.g., via the normalized Hermite polynomials), the authors propose to assess the smoothness by studying the corresponding matrix-valued kernel.

From my personal viewpoint, the major contribution of this paper is:
1. propose to study the smoothness properties of vector Gaussian processes on product space via a spectral approach; and
2. derive precise smoothness characterizations in Section 3.4.

**Audience:**

Yes

**Broader Impact Concerns:**

This paper is primarily theoretical and I do not see any ethical concerns.

**Claims And Evidence:**

Yes

**Requested Changes:**

1. why the setting of product space between unit sphere and standard Euclidean space? The authors mentioned on possible applications at the end of Section 1.2, but the connections are a bit vague and unclear. In fact, it could be to good idea to showcase the application of the proposed characterization on a concrete ML example.
2. it would be helpful to point to, when talking about the contribution of this work at the top of P3, the corresponding precise results in the paper. For example, the route is given in P5, the spectral characterization of smoothness that relies on the properties of the matrix-valued kernel is given in xxx, etc.
3. Please refer to the precise section of proof in the Appendix.

A few minor issues:
1. Corollary 3.4: "one of the following measureS"?
2. what does $\ell^2$ mean in Proposition 3.3?

**Strengths And Weaknesses:**

Strengths: the paper is primarily theoretical and the theoretical results are, to the best of my knowledge, novel and interesting.

Weaknesses:
1. The motivation on the study of smoothness properties, and the specific choice of product is a bit vague and unclear; e.g., more connection to applications may be helpful.
2. The paper is rather technical and the presentation can be improved.
See my detailed comments below.

---

### Review · Reviewer_wYMA · 2024-03-17

**Summary Of Contributions:**

This paper addresses the issue of quantifying the smoothness of vector Gaussian processes defined over the product space $\mathbb S^d \times \mathbb R^k$. The paper considers a class of kernels over this space which are isotropic and stationary, i.e., given inputs $(x, t)$, $(x’, t’)$, the output of the kernel only depends on $\langle x,x’\rangle$ and $t-t’$. For such kernels, by leveraging the expansion in the orthonormal basis of corresponding $L^2$ space (induced by the product measure of the spherical Haar measure and the Gaussian measure), the paper relates the decay of the Fourier coefficients in this basis with appropriate Sobolev-type norms. The flavor of the result is expected and the main contribution seems to be to write everything down explicitly.

**Audience:**

Yes

**Broader Impact Concerns:**

None.

**Claims And Evidence:**

Yes

**Requested Changes:**

The notation $\lVert\cdot\rVert_k$ is usually used for the $\ell_k$ norm, and $\lVert\cdot\rVert_*$ is usually used for the nuclear norm; please change these notations (it would be fine to use $\lVert\cdot\rVert$ for both, or $\lVert\cdot\rVert_{\rm F}$ or $\lVert\cdot\rVert_{\rm HS}$ for the Frobenius norm).

Please make Section 2 more self-contained by formally stating and explaining the application of formula 11.4.2 from Erdélyi (1953) and Cramér's theorem within the context of your analysis.

Typos:
- Section 1.2: change the title from "Why Studying" to "Why Study".
- In eq. (5), the last comma should be a period.
- After eq. (10): "indexes" -> "indices".

**Strengths And Weaknesses:**

Strengths:

The spectral analysis is well-founded and the paper is carefully written.

Weaknesses:

The choice of notation, such as using $\lVert\cdot\rVert_k$ for the Euclidean norm and $\lVert\cdot\rVert_*$ for the Frobenius norm, might confuse readers familiar with the conventional usage of these notations in other contexts. Also, the reliance on external references, such as Erdélyi (1953) and Cramér's theorem, without fully integrating their details into the discussion, detracts from the paper's self-contained nature and accessibility.

---

> ### Author Response · Authors · 2024-03-27
> **Replies**
>
> We are extremely grateful to the Editors and the Referees for their great job that allowed for a considerably improved version of the manuscript.
> A carefully revised version, in concert with a Supplement that answers all the points by the Referees, are attached to this second submission.
> Thank you, best wishes,
> The Authors

---

### Review · Reviewer_TRSJ · 2024-03-20

**Summary Of Contributions:**

This paper characterizes the smoothness properties of vector Gaussian processes. The authors first use Hermite polynomials to construct a spectral decomposition of a matrix-valued kernel, then define a corresponding Sobolev space, then quantify the smoothness of a given kernel by investigating the $s_{j,\beta}$ functions in (14). The main result is in proposition 3.3, stating that smoothness is equivalent to summability conditions of the matrix coefficients of the spectral decomposition.

**Audience:**

Yes

**Broader Impact Concerns:**

No ethical implications were identified.

**Claims And Evidence:**

Yes

**Requested Changes:**

Somewhere around sec 2, the authors should explain how smoothness is quantified for scalar Gaussian processes, so the readers can see the developments are not trivial. Overall, this paper introduces a construction but does not compare it with alternative constructions.

How significant is the studied product space? Can either the sphere or the Euclidean space be zero dimension (d=0 or k=0), so that the product space includes the sphere/Euclidean space as special cases?

1.4 introduce \mathbb{Z}

P4 "cov" is not introduced

3.1 "It can be namely verified that these functions ..." Add more explanations and intuition.

3.3. Explain why $(-1,1) \times R^k$ is investigated. Why the interval (-1,1) is not closed?

sec 3. In the end, how smoothness is "quantified"? Is it a binary property (whether or not K belongs to the space $W_{d,k}^{\zeta,m}$)?

**Strengths And Weaknesses:**

Strength:
Novel construction to define smoothness of vector-valued Gaussian process.

Weaknesses:
The construction depends on some recent references, e.g. Porcu et al. 2021. And therefore it is not introduced in a self-contained way.

---

### Decision · Action_Editor_54Nj · 2024-05-01

**Recommendation:** Accept as is

**Comment:**

The paper is primarily theoretical and the theoretical results seem to be novel and interesting.

Strengths:
- Novel construction to define smoothness of vector-valued Gaussian process.
- The spectral analysis is well-founded and the paper is carefully written.
- Derivation of precise smoothness characterizations in Section 3.4.
- The presentation is relatively clear, with the "route to smoothness" helping the readers understand the proof process, despite the technicalities.
- The logic is presented clearly, and the arguments are mathematically sound.

Virtually all weaknesses have been addressed after a revision.

**Audience:**

Theoreticians working in the area of vector Gaussian processes

**Claims And Evidence:**

This paper characterizes the smoothness properties of vector Gaussian processes. The authors first use Hermite polynomials to construct a spectral decomposition of a matrix-valued kernel, then define a corresponding Sobolev space, and subsequently quantify the smoothness of a given kernel by investigating the $s_{j,\beta}$ functions in (14). The main result is in Prop 3.3, stating that smoothness is equivalent to summability conditions of the matrix coefficients of the spectral decomposition.